# In Vitro Evaluation of Surface Roughness and Color Variation after Two Brushing Protocols with Toothpastes Containing Different Whitening Technologies

Angel Lobito [1,*], Catarina Colaço [1], Joana Costa [1], Jorge Caldeira [1,2], Luís Proença [1]
and José João Mendes [1]

1   Clinical Research Unit (CRU), Egas Moniz Center for Interdisciplinary Research (CiiEM), Egas Moniz School
    of Health & Science, 2829-511 Almada, Portugal; ccolaco@egasmoniz.edu.pt (C.C.);
    jcosta@egasmoniz.edu.pt (J.C.); jcaldeira@egasmoniz.edu.pt (J.C.); lproenca@egasmoniz.edu.pt (L.P.);
    jmendes@egasmoniz.edu.pt (J.J.M.)
2   LAQV Requimte, Faculdade de Ciências e Tecnologias, Universidade Nova de Lisboa,
    2829-516 Almada, Portugal
*   Correspondence: alobito@egasmoniz.edu.pt

**Abstract:** The aim was to evaluate the effect of different whitening toothpastes on the enamel surface roughness and color variation. Twenty-four molars were sectioned and divided into eight groups (*n* = 3) considering the following two factors under study: toothpaste type (Colgate® Total Original, Oral B® 3D White Luxe Perfection, Curaprox® Black is White, and Signal® White Now) and brushing protocol (short- and long-term). Surface roughness was examined by atomic force microscopy (AFM), and color change (ΔE) was measured using the CIE L*a*b* system. Data were statistically analyzed using comparative parametric tests at a 5% significance level. In the short-term protocol, only the Signal® White Now toothpaste increased surface roughness (*p* = 0.038) compared to the Colgate® Total Original group. No significant differences (*p* > 0.05) were observed in surface roughness in the long-term protocol. Regarding color variation, no statistically significant differences (*p* > 0.05) were observed in either protocol. Overall, the whitening toothpastes did not affect enamel surface roughness or color, except for Signal® White Now, which caused increased roughness in the short-term protocol. However, all toothpastes induced a visual change in color.

**Keywords:** roughness; color; enamel; whitening; toothpaste; brushing; aesthetics

## 1. Introduction

Achieving an aesthetically pleasing smile often holds greater significance for patients than prioritizing their oral health [1,2]. Consequently, a considerable number of individuals opt for over-the-counter whitening toothpastes, often without a comprehensive understanding of associated risks or its appropriate usage protocols [3].

Tooth discoloration represents a prevalent concern among the population; its origins derive often from dental pigmentation, characterized by the deposition of pigments within the tooth structure. The pigmentation can be of intrinsic and/or extrinsic origin, the latter being the most predominant type that can be removed prophylactically through daily oral hygiene practices and with the use of toothpastes. This discoloration occurs when chromogenic substances adhere to the enamel surface and change its original color [4,5].

Whitening toothpastes have been observed to effectively remove and manage the deposition of chromophores responsible for extrinsic pigmentation [6–9] and the color of teeth is altered by increasing the brightness of the tooth structure, through a combination of mechanical and chemical reactions [10]. These toothpastes commonly contain abrasive particles, optical agents, and/or chemical agents [11].

Whitening toothpastes incorporating abrasive microspheres rely on a mechanical mechanism for pigment removal targeting biofilm and extrinsic pigmentation. Among

the abrasives commonly employed in these formulations are calcium carbonate, hydrated silica, dicalcium phosphate, aluminum oxide, and sodium bicarbonate [8,9,12–15].

Activated charcoal toothpastes typically incorporate various abrasives, with silica as the most prevalent. The more abrasive the formulation, the more effective it is at removing extrinsic pigmentation; however, prolonged use may lead to the removal of tooth surface, which can lead to changes in surface roughness [12,13]. Activated charcoal operates by binding to the deposits, bacteria, and pigments (hydrophobic) present on the tooth surface, facilitated by its high surface area, porous nature, and high hydrophobicity. This high adsorption capacity can counteract the effects of fluoride ions, which are typically present in lower concentrations (950 ppm) compared to those found in conventional toothpaste formulations [9,12].

Recently introduced to the market, blue covarine-based whitening toothpastes incorporate a whitening agent that has an optical effect on the color of the tooth surface, altering its perceived shade [7,8]. This optical operates by altering the apparent color of the tooth surface through a uniform deposition of a thin, semi-transparent layer of bluish pigment uniformly on it, with the intensity of the effect correlating with pigment concentration. Examining the additional constituents of whitening toothpastes that include blue covarine reveals the presence of modified silica particles that have an abrasive action on enamel. This coating immediately modifies the interaction and perception of incident light, which is an advantage of this agent and results in yellowed teeth that convey the effect of being whiter and brighter due to the change in the b* axis of the CIE L*a*b* [7,8,14].

In 2019, Vaz et al. conducted a study to evaluate a range of whitening toothpastes with different mechanisms of action, including activated charcoal, blue covarine, hydrogen peroxide, and microbeads [8]. However, this study was limited to the evaluation of color and used a visual assessment method, which is highly subjective. Visual assessments of color variation have a high risk of bias and cannot accurately correlate to the in vivo performance. In an attempt to achieve a more realistic assessment, in our study a colorimeter (Optishade Style Italiano, Smile Line, St-Imier, Switzerland) was used. However, there is a lack of scientific support for this type of instrument [16].

While the efficacy of the whitening toothpastes is often only evaluated based on color changes alone [8], there is a notable gap in understanding their effects on enamel surface roughness. A systematic review was performed in 2022, assessing the effect of whitening toothpaste on the surface roughness of human teeth, including seven studies from which four were included in a meta-analysis [17]. Therefore, more studies are needed to perform a systematic review with more power to evaluate multiple parameters [18].

Understanding the impact of these toothpastes on enamel roughness and color appearance is crucial for offering both professionals and patients valuable insights into balancing desired dental aesthetics with the maintenance of tooth structural integrity. Thus, the objective of this study was to assess the influence of various whitening toothpastes containing different active agents on enamel surface roughness and color variation. This investigation involved two parallel studies, evaluating the effects over both short- and long-term periods using distinct mechanical brushing protocols.

## 2. Materials and Methods

### 2.1. Tooth Preparation

This in vitro experimental study was performed on twenty-four permanent human molars, without caries or any type of restoration, obtained from the biobank of the Egas Moniz Dental Clinic (approved by the Ethics Committee of the Egas Moniz School of Health and Science, Portugal, n° 1142).

All teeth were cleaned in running water with detergent with the help of a brush/sponge, followed by scraping with a specific Gracey curette for posterior teeth, and an HW-3H scaler (Woodpecker, Guilin, China). Subsequently, polishing of the teeth was carried out with a prophylaxis paste (Henry Schein, Melville, NY, USA) and pumice stone powder in a counter-angle with a prophylactic brush. After this procedure, the teeth were placed in a

0.5% chloramine trihydrate (*v*/*v*) solution for one week and then stored in distilled water in a refrigerator at 4 °C (Bosch GmbH, Munich, Germany) and changed weekly [19].

Twenty-four teeth were sectioned in a mesial–distal direction with a single cut, using a hard tissue microtome (Accutom-50, Struers A/S, Ballerup, Denmark), at a cutting speed of 0.350 mm/s and a rotation of 3200 rpm while irrigated with deionized water. With the implementation of this step in the protocol, it was possible to duplicate the sample number by means of obtaining two identical surfaces per tooth sectioned at the buccal and lingual/palatal sections. Pulp remnants were removed from the internal surface and the pulp cavity was sealed with cyanoacrylate glue (Loctite, Bilbao, Barcelona, Spain). Of the forty-eight surfaces obtained, half were used to assess surface roughness and the other half for color variation. Both assessments were classified as independent studies to evaluate the potential interactions between the two independent variables under study (*n* = 3): toothpaste types and brushing protocol duration.

The twenty-four specimens used for surface roughness evaluation, within each group, were immersed in 13 mL of deionized water in a single sterilized tube (VWR, Matsonford, PA, USA) in a refrigerator (Bosch GmbH, Munich, Germany) at 4 °C, where they remained until the first roughness evaluation. As of the start of the brushing protocols, all specimens were transferred to an incubator at 37 °C (Memmert, Schwabach, Germany) and immersed in artificial saliva (Table A1, Appendix A).

The remaining twenty-four specimens used to evaluate surface color variation were subjected to a staining protocol that consisted of the placement of each specimen in a 15 mL single sterilized tube (VWR, Matsonford, PA, USA), containing a concentrated coffee solution prepared by mixing 120 mL of boiling water with 2.4 g of instant coffee (Delta Cafés, Campo Maior, Portugal), so that each specimen remained immersed in 5 mL of the solution [20]. The staining protocol consisted of four cycles of 18 h immersion in the staining solution followed by 6 h of drying [8,21]. Throughout the cycles, the samples remained in the incubator at 37 °C (Memmert, Schwabach, Germany). After the staining protocol and before the initial color evaluation, the samples remained in the same conditions as the specimens used for the surface roughness evaluation.

## 2.2. Brushing Protocols

In both the surface roughness and color variation studies, the specimens were randomly divided into eight different experimental groups based on potential interactions between the two independent variables studies (*n* = 3): toothpaste types and brushing protocol duration.

Within each protocol, four groups were assigned to different toothpastes:

- Control group (CTO): Colgate® Total Original (Colgate, Palmolive, Porto Salvo, Portugal) conventional toothpaste;
- Group 1 (OB3D): Oral B® 3D White Luxe Perfection (Procter & Gamble, Schwalbach am Taunus, Germany) based on abrasive microsphere whitening technology;
- Group 2 (CBW): Curaprox® Black is White (Curaden Swiss Headquarters, Kriens, Switzerland) based on the activated charcoal whitening technology;
- Group 3 (SWN): Signal® White Now (Unilever RA, Rueil-Malmaison, France) based on blue covarine whitening technology.

These four types of toothpaste (one control and three whitening) share several ingredients with each other (Table A2, Appendix A).

Two different brushing protocols (S and L) were designed to obtain short- and long-term results. Protocol S aimed to mimic human behavior by performing one cycle (5 s), three times a day for 15 days. Protocol L replicated intense brushing, consisting of 30 cycles, 6 times a day, for 3 days, simulating the number of cycles the tooth surface would receive if brushed one cycle three times a day for 6 months, totaling 540 cycles. Both of the mechanical brushing protocols S and L were each carried out with an electric toothbrush Oral B Pro 3 3700 (Braun GmbH, Frankfurt, Germany) with a pressure sensor, by the same pre-calibrated investigator (A.L.), applying a defined volume of toothpaste (6 × 3 × 2 mm)

on the respective Oral B CrossAction brush head with soft bristles (Braun GmbH, Frankfurt, Germany), exclusive for each of the toothpaste group.

After each brushing, the samples were rinsed with deionized water for 10 s and restored in artificial saliva (Table A1, Appendix A), which was renewed daily, and then placed back in the incubator at 37 °C after each brushing procedure (Memmert, Schwabach, Germany).

### 2.3. Surface Roughness Measurement

Surface roughness measurements (Ra) were performed before the initial brushing and after the final one, using an atomic force microscope (AFM) TT-AFM (AFM Workshop, Signal Hill, CA, USA), with the following formula [22,23]:

$$\text{Ra} = \frac{1}{L}\int_0^L |Z(x)|\,dx \tag{1}$$

The deflection and height-mode images of the samples were obtained with a scan rate of 0.7 Hz, using a vibration mode with a resolution of $512 \times 512$ pixels. Within each sample, a region with dimensions of $40 \times 40$ μm was randomly selected, which, using the Gwyddion 2.63 software (CMI, Brun, Czech Republic), made it possible to obtain 16 observations of $10 \times 10$ μm, resulting in a total of 48 observations for each group in each protocol (under ideal conditions). Since the AFM is limited to a maximum height variation of 17 μm in the area to be evaluated, and taking into account the irregularity of the enamel surface, there were cases in which it was necessary to 'reject' some of the zones obtained because they had atypical values that did not represent the real condition of the enamel. The Ra values were recorded for all the areas tested, representing the average roughness value, and then the difference between the average final and initial roughness values was carried out.

### 2.4. Color Variation

Color measurements were conducted at two distinct time points: the initial color, immediately following the staining procedure, and the final color, after the completion of the brushing protocols. These measurements were performed with a colorimeter (Optishade Style Italiano, Smile Line, St-Imier, Switzerland) on the tooth crown surface in three different zones: occlusal, middle and cervical [24,25]. This instrument registered the parameters of the CIE L*a*b* system, and the total overall color change (ΔE) was calculated using the formula [7,8,14,26]:

$$\Delta E = \left[ (\Delta L^*)^2 + (\Delta a^*)^2 + (\Delta b^*)^2 \right]^{1/2} \tag{2}$$

The National Institute of Standards and Technology recommends converting ΔE to National Bureau of Standards (NBS) units by applying the equation:

$$\text{NBS units} = \Delta E \times 0.92 \tag{3}$$

to assess the color differences as shown in Table 1 [26–28].

**Table 1.** National Bureau of Standards (NBS) units for expressing color differences.

| NBS units | Color Differences |
| --- | --- |
| <0.5 | Extremely slight change |
| 0.5–1.5 | Slight change |
| 1.5–3.0 | Perceivable change |
| 3.0–6.0 | Marked change |
| 6.0–12 | Extremely marked change |
| ≥12 | Change to another color |

*2.5. Statistical Analysis*

The statistical analysis program IBM SPSS Statistics version 29.0 (IBM, Armonk, NY, USA) was used to analyze all data obtained in this research, using descriptive and inferential statistical analysis methodologies.

Since normality and homogeneity of variance were verified (Shapiro–Wilk and Levene tests, $p > 0.05$), a one-way ANOVA with post hoc Tukey HSD parametric tests was used. In all statistical tests, the level of significance was set at 5%.

## 3. Results

*3.1. Surface Roughness*

Descriptive analysis was performed with the mean and standard deviation values of surface roughness difference shown in Table 2. Higher surface roughness values were obtained in the SWN group in protocol S with an increase of 436.2 ($\pm$ 141.8) nm. Conversely, lower values were obtained in the CTO group in the same protocol with a decrease of 66.1 ($\pm$ 139.0) nm.

**Table 2.** Distribution of surface roughness differences ($\Delta$Ra, nm) presented as mean ($\pm$ standard deviation, SD) among the groups, according to the experimental protocol ($n$ = 3).

| $\Delta$Ra (nm) M ($\pm$ SD) | Protocol S | Protocol L |
|---|---|---|
| CTO | −66.1 ($\pm$ 139.0) [a] | −19.9 ($\pm$ 301.3) [a] |
| OB3D | 150.4 ($\pm$ 130.8) [a] | −51.2 ($\pm$ 4.5) [a] |
| CBW | 400.7 ($\pm$ 273.5) [a] | 115.5 ($\pm$ 259.0) [a] |
| SWN | 436.2 ($\pm$ 141.8) [b] | 155.2 ($\pm$ 123.0) [a] |

M = mean, SD = standard deviation. Different lowercase letters indicate significant differences between means in the same protocol (Tukey HSD post hoc test, $p < 0.05$).

In protocol S, there were statistically significant differences in the surface roughness difference mean values among the four different toothpaste groups ($p$ = 0.03, ANOVA). The SWN toothpaste resulted in significantly ($p$ = 0.038, Tukey HSD) higher surface roughness differences compared to the control group (CTO group). In the other groups (OB3D and CBW), the surface roughness differences' mean values were not statistically significant compared to the control group ($p > 0.05$, Tukey HSD).

On the other hand, in protocol L, no statistically significant differences were observed among the surface roughness difference mean values from the toothpaste experimental groups ($p$ = 0.576, ANOVA).

*3.2. Color Variation*

Descriptive analysis was performed with the mean and standard deviation values of color variation shown in Table 3. Color difference values ranged between 14.9 ($\pm$ 2.4), in the OB3D group in protocol S, and 10.7 ($\pm$ 2.9) in the CTO group in protocol L.

**Table 3.** Distribution of color variation ($\Delta$E) presented as mean ($\pm$ standard deviation, SD) among the groups, according to the experimental protocol ($n$ = 3).

| $\Delta$E M ($\pm$ SD) | Protocol S | Protocol L |
|---|---|---|
| CTO | 14.7 ($\pm$ 3.7) | 10.7 ($\pm$ 2.9) |
| OB3D | 14.9 ($\pm$ 2.4) | 10.9 ($\pm$ 2.5) |
| CBW | 11.6 ($\pm$ 3.2) | 12.2 ($\pm$ 2.4) |
| SWN | 11.8 ($\pm$ 1.1) | 11.7 ($\pm$ 3.9) |

M = mean, SD = standard deviation.

For both protocols, there were no statistically significant differences in the color difference values among the four different toothpaste groups ($p > 0.05$, ANOVA).

Table 4 shows the data for NBS. In protocol S, the CTO and OB3D caused a change to another tooth color. The other two groups of protocol S and all four experimental groups of protocol L caused an extremely marked color change to the teeth.

**Table 4.** Distribution of color perception (NBS units, presented as absolute value, and color differences) among the groups, according to the experimental protocol ($n$ = 3).

| NBS Units Color Differences | Protocol S | Protocol L |
|---|---|---|
| CTO | 13.5 Change to another color | 6.7 Extremely marked change |
| OB3D | 13.7 Change to another color | 7.5 Extremely marked change |
| CBW | 10.7 Extremely marked change | 11.2 Extremely marked change |
| SWN | 10.8 Extremely marked change | 10.8 Extremely marked change |

NBS = National Bureau of Standards.

## 4. Discussion

One of the most critical aspects of an aesthetically pleasing smile is the color of the teeth, which can affect the patient's self-esteem, social interactions, environmental adaptations, employment opportunities and other important aspects that affect their quality of life. Therefore, whitening toothpastes have gained popularity due to their convenience use and widespread accessibility [29].

An ideal whitening toothpaste should effectively remove extrinsic stains while causing minimal effect on tooth structure, but patients are driven by social media and marketing strategies to purchase whitening toothpastes as a reliable and effective alternative to more efficient but more expensive treatments in order to achieve whiter teeth without really understanding their effects. It is important to clarify that these types of whitening toothpastes lack scientific evidence, and their results are highly manipulated by the industry. Further research with robust methodology that reduces the risk of bias is needed to establish the role of abrasive and whitening components present in these toothpastes and their correlation with alterations in surface roughness and color [6,30].

It is important to understand the effects of the different active agents present in whitening toothpastes and their effect on the enamel surface roughness and color following the application of two different mechanical brushing protocols. Such insights are essential to guide both oral health professionals and patients in the selection of an appropriate whitening toothpaste, balancing the desired aesthetic goals with the preservation of the structural integrity of the teeth.

Surface roughness plays a critical role in the retention and adhesion of substances such as bacteria and pigments to the enamel surface. According to the literature, a surface with an average roughness greater than 200 nm is more likely to accumulate bacteria and pigments, which in turn leads to an increase in bacterial plaque, thereby increasing the risk of dental caries, periodontal disease, and tooth discoloration. In addition, enamel surface roughness is a critical variable, as it can affect not only the aesthetic aspects of the smile, but also the enamel's resistance to erosive processes [31–34].

In protocol S of the present study, only the SWN toothpaste increased the surface roughness values compared to the control group (CTO). This can be explained due to the abrasive components present in the composition of the toothpaste. The manufacturer does not specify the exact amount of the different abrasives, nor the relative dentin/enamel abrasiveness. Having high abrasiveness values could be one of the reasons for the statistically significant values in surface roughness. An experimental technique to quantify the abrasive content of different toothpastes requires further studies. Using a combination of scanning

electron microscopy (SEM) and energy-dispersive X-ray spectroscopy (EDS) could aid in the identification of the composition of the abrasive particles and their concentration [35,36]. While the study conducted by Shamel et al. (2019) [27] did not verify this, it is plausible that the methodological choice of mixing the toothpaste with distilled water could have influenced the results.

In the same protocol, OB3D and CBW toothpastes did not show differences in surface roughness. These findings are in line with other studies, such as those by Yaghini et al. (2023) and Shaikh et al. (2021) that also concluded that the use of charcoal whitening toothpastes did not increase enamel surface roughness [18,37]. However, there are differences between the methodology of the studies, with the main one being the usage of bovine teeth in the studies carried out by Shaikh et al. (2021) [37].

Despite this result, the CBW group displayed a tendency to higher average roughness values in comparison to the control group. The observed decrease in values within the control group across both protocols can be attributed to the absence of whitening agents in this toothpaste variant (control group). This validates the study's findings.

In the long-term protocol, none of the three whitening toothpastes resulted in differences in the enamel surface roughness. However, it is noteworthy that the surface roughness value recorded for the OB3D group in protocol L differs from the value reported in the current literature [10], which may have resulted from the inherent limitations of an in vitro study.

Regarding the color evaluation for protocol S and considering the NBS scale, no differences were observed within the groups. However, the CBW and SWN groups exhibited a "extremely marked change", while the CTO and OB3D groups demonstrated a "change to another color". Nevertheless, it would be expected that the control group, which is absent of whitening agents, would exhibit lower scores than any of the other groups. This can be attributed to the fact that this toothpaste, although considered "conventional" and containing no additional whitening agent, also contains abrasive particles that remove extrinsic pigmentation from tooth surfaces. Additionally, the literature suggests that the efficacy of extrinsic pigment removal is also directly related to the level of oral hygiene practices, motivation, and the mechanical action of the toothbrush, which may influence the results regardless of the toothpaste used [30,38]. The outcome was as expected for the three whitening toothpastes, all of which demonstrated a whitening effect on the enamel surface, probably due to the chemical and mechanical combination of the whitening and abrasive agents, respectively.

In the long-term protocol and considering the NBS score, all groups registered a score between 6 and 12, which is considered a "extremely marked change". The control group had the lowest score, followed by the OB3D, SWN, and CBW groups, contrary to what was expected based on the assumption that the effects registered in the S protocol would only intensify in the L protocol [10].

Protocol S was designed to mimic normal daily brushing of three cycles per day for 15 days. Protocol L aimed to simulate, in just three days, the same number of cycles that would be obtained if the samples were brushed three times a day for six months, to try and simulate the long-term effects of different toothpastes. Regarding the results of protocol L, one would anticipate the values to align or potentially exacerbate those recorded in protocol S, given that the aim of protocol L was to simulate extensive use. However, it is important to note that this discrepancy was not the primary focus of the study, nor was it subjected to statistical analysis. We can assume that for protocol L, within each of the 30 brushing cycles, the amount of toothpaste used replicated only the amount used in a single brushing, which may have caused a loss in properties over time within the same cycle, due to the long duration.

The Optishade Style Italiano is a contemporary and underutilized colorimeter, with only one scientific study existing in the field of endodontics [39]. Equipped with red, green, and blue filters to approximate the human eye's spectrum, this colorimeter operates by capturing colors through processing the light reflection via said filters [40].

In every in vitro study, inherent limitations are present during its execution. In this study, the limitations include the size of the sample and the irregularity of the enamel surface, which interact with the small size of the area studied (40 μm × 40 μm × 17 μm). The brushing mechanism utilized also posed a limitation, as it was not possible to employ an automatic brushing apparatus. Instead, an electric toothbrush was used, which hindered the ability to predict and rigorously control various factors such as temperature, revolutions per minute, and the pressure exerted between the different samples. Other limitations may derive from the operator such as fatigue, stress, the angle of the brush bristles, and excessive pressure. Furthermore, inherent flaws in analytical instruments, such as the nanometric probes utilized in the AFM and the calibration card of the colorimeter, may contribute to inaccurate measurements, thereby representing an additional limitation in this study.

Further studies with microhardness, SEM images, and EDS, for example, should be performed to enhance the comprehensiveness of this information and facilitate better clinical application.

## 5. Conclusions

According to the results obtained and attending to all the limitations associated with an in vitro laboratory study, it was possible to conclude that the different whitening toothpastes did not affect the surface roughness of enamel, either in short- or long-term protocols. However, some caution should be taken regarding toothpastes with blue covarine whitening technology, which showed the highest increase in surface roughness.

In terms of color variation after pigmentation with the coffee solution, and, in accordance with the limitations mentioned above, the whitening toothpastes were effective in changing the color of the tooth surface of the samples studied. However, there were no variations between the four toothpastes, so all were effective in removing extrinsic pigmentation.

Whitening toothpastes are increasingly being used by patients as a quick, inexpensive, and effective solution to remove unwanted pigmentation. However, there remains a lack of studies conducted under comparable conditions for the three whitening technologies examined in this investigation. Dentists play a pivotal role in acquiring current evidence on this subject to confidently recommend suitable whitening toothpaste options and offer well-informed guidance to patients in selecting the most appropriate tooth whitening product, considering not only their efficacy but also to avoid oral health complications.

**Author Contributions:** Conceptualization, A.L., C.C., J.C. (Joana Costa) and J.J.M.; methodology, A.L. and C.C.; validation, J.C. (Joana Costa) and J.J.M.; investigation, A.L., C.C., J.C. (Joana Costa) and J.C. (Jorge Caldeira); resources, J.J.M.; statistical analysis, J.C. (Joana Costa) and L.P.; data curation, A.L., C.C., J.C. (Joana Costa) and L.P.; writing—original draft preparation, A.L., C.C. and J.C. (Joana Costa); writing—review and editing, J.C. (Joana Costa), J.C. (Jorge Caldeira) and L.P.; supervision, J.J.M. All authors have read and agreed to the published version of the manuscript.

**Funding:** This research was funded by CiiEM Investiga 2019 Shining Spectroscopic Smile El19/12.

**Institutional Review Board Statement:** The study was approved by the Ethics Committee of the Egas Moniz School of Health & Science, Portugal (n° 1142 and date of approval 15/12/2022).

**Informed Consent Statement:** Not applicable.

**Data Availability Statement:** The original contributions presented in the study are included in the article and further inquiries can be directed to the corresponding authors.

**Conflicts of Interest:** The authors declare no conflicts of interest.

## Appendix A

**Table A1.** Composition of artificial saliva (with respective quantification).

| Compound | Quantity |
|---|---|
| NaCl | 0.80 g |
| KCl | 0.80 g |
| $CaCl_2 \bullet 2H_2O$ | 1.812 g |
| $NaH_2PO_4 \bullet 2H_2O$ | 1.38 g |
| $Na_2S \bullet 9H_2O$ | 0.01 g |
| Urea | 2 g |
| Distilled $H_2O$ | 2000 mL |

**Table A2.** Toothpastes used in this study.

| Toothpaste Name | Tooth Technology | Composition | Manufacturer (Batch Number) |
|---|---|---|---|
| Colgate® Total Original (CTO) | Control non-whitening | Glycerin, Aqua, Hydrated Silica, Sodium Lauryl Sulfate, Arginine, Aroma, Cellulose Gum, Zinc Oxide, Poloxamer 407, Zinc Citrate, Tetrasodium Pyrophosphate, Xanthan Gum, Benzyl Alcohol, Cocamidopropyl Betaine, Sodium Fluoride (1450 ppm F⁻), Sodium Saccharin, Phosphoric Acid, Hydroxypropyl Methylcellulose, Sucralose, CI 73360, CI 74160, CI 77891. | Colgate, Palmolive, Porto Salvo, Portugal (3104PL1171) |
| Oral B® 3D White Luxe Perfection (OB3D) | Abrasive microsphere whitening technology | Glycerin, Hydrated Silica, Sodium Hexametaphosphate, Aqua, PEG-6, Aroma, Trisodium Phosphate, Sodium Lauryl Sulfate, Cocamidopropyl Betaine, Sodium Saccharin, Sodium Fluoride (1450 ppm F⁻), Carrageenan, PVP, Xanthan Gum, Limonene, Sucralose, Sodium Benzoate, Sodium Hydroxide, CI 74160, Citric Acid, Sodium Citrate, Potassium Sorbate. | Procter & Gamble, Schwalbach am Taunus, Germany (2306G7) |
| Curaprox® Black is White (CBW) | Activated charcoal whitening technology | Aqua, Sorbitol, Hydrated Silica, Glycerin, Charcoal Powder, Aroma, Decyl Glucoside, Cocamidopropyl, Betaine, Sodium Monofluorophospate (950 ppm F⁻), Tocopherol, Xanthan Gym, Maltodextrin, Mica, Hydroxyapatite (Nano), Potassium Acesulfame, Titanium Dioxide, Micro-Crystalline Cellulose, Sodium Chloride, Potassium Chloride, Citrus Limon Peel Oil, Sodium Hydroxide, Zea Mays Starch, Amyloglucosidase, Glucose Oxidase, Urtica Dioica Leaf Extract, Potassium Thiocyanate, Cetearyl Alchohol, Hydrogenated Lecithin, Menthyl Lactate, Mehtyl Diisopropyl Propionamide, Ethyl Menthane Carboxamide, Stearic Acid, Mannitol, Sodium Bisulfite, Tin Oxide, Lactoperoxidase, Limonene. | Curaden Swiss Headquarters, Kriens, Switzerland (199MHD) |
| Signal® White Now (SWN) | Blue covarine whitening technology | Aqua, Hydrogenated Starch Hydrolysate, Hydrated Silica, Sodium Lauryl Sulfate, Aroma, Cellulose Gum, Sodium Saccharin, Sodium Fluoride (1450 ppm F⁻), PVM/MA Copolymer, Glycerin, CI 42090, CI 74160. | Unilever RA, Rueil-Malmaison, France (2038FCA) |

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
