# Peer review of "In Vitro Evaluation of Surface Roughness and Color Variation after Two Brushing Protocols with Toothpastes Containing Different Whitening Technologies"

_applsci, doi:10.3390/app14104053_

Round 1
Reviewer 1 Report
Comments and Suggestions for Authors
Line 175
Please, can you explain which region of the tooth surface was detected for colour measurements, if the same position was detected again after the brushing protocols or if the hole surface was considered for colour evaluation? And how the average on the hole surface was done to avoid overlapping?
Additional comments:
Regarding the methodology used in the study, the authors conducted two types of measurements - Surface Roughness Measurement and Color Variation. The first experiment involved selecting a random region on the surface of the samples and obtaining 16 observations for each sample, which is described in detail by the authors. The final value is the average difference between the final and initial values.
The authors used a dental colorimeter (Optishade Style Italiano) to measure color variation, but they did not specify which sample's region was used for the measurement. It is unclear whether the color measurement was taken from the entire surface of each sample or from a specific region that was evaluated before and after the use of the whitening toothpaste.
In my opinion, the conclusions are coherent and have addressed the main question.
Author Response
Dear reviewer,
Thank you very much for taking the time to review this manuscript and we appreciate the opportunity to clarify and improve the manuscript in line with your suggestions. Hereby the authors present a point-by-point response to all the comments and suggestions kindly given by you.
The corresponding revisions are marked in colored text (yellow) in the manuscript file (version R1). New references were added.
We hope the answers below and modifications introduced in the manuscript are clear and concise enough and that the manuscript now matches the high standards of the “New Techniques, Materials and Technologies in Dentistry: Second Edition”, a special issue of Applied Sciences that belongs to the section "Applied Dentistry and Oral Sciences”.
We deeply appreciate the insightful feedback provided by the reviewer in the materials and methods section. Their constructive criticism has significantly enhanced the clarity and robustness of the described methodologies. Moreover, their suggestions facilitated the integration of pertinent references, thereby augmenting the comprehensiveness and validity of the manuscript under review.
Comment 1: Please, can you explain which region of the tooth surface was detected for colour measurements, if the same position was detected again after the brushing protocols or if the hole surface was considered for colour evaluation? And how the average on the hole surface was done to avoid overlapping?
Response 1: The authors appreciate the reviewer’s comment and agree that the region of color measurement on the tooth surface warranted clear elucidation in the initial article submitted. To ensure consistency in color measurements, the teeth were systematically divided into three distinct zones: occlusal, middle, and cervical. The average values of L, a, and b parameters were computed from measurements obtained in these delineated areas. This approach facilitated the maintenance of consistent measurement positions throughout the study, both before and after the brushing protocols were conducted.
This protocol was based on some articles, such as
- Chu, S. J.; Trushkowsky, R. D.; Paravina, R. D. Dental color matching instruments and systems. Review of clinical and research aspects. Journal of dentistry 2010, 38 Suppl 2, e2–e16. doi:10.1016/j.jdent.2010.07.001
- RutkÅ«nas, V.; DirsÄ—, J.; Bilius, V. Accuracy of an Intraoral Digital Scanner in Tooth Color Determination. The Journal of Prosthetic Dentistry 2020, 123, 322–329, doi:10.1016/j.prosdent.2018.12.020.
In response to this, a paragraph has been included in the revised manuscript (line 179-180) to elaborate on these details.

Reviewer 2 Report
Comments and Suggestions for Authors
Dear authors,
The manuscript was well written, it is well aligned with the scope of the journal, and it can reach a broad audience and readership. Although the experimental techniques were very limited (only color change and AFM), the results were contextualized with the literature, included proper statistical treatment and were presented in a compelling way.
I only have a few comments/questions:
1) Please include the formula for the Surface roughness measurement (Ra) (line 161). The authors did include for delta E (line 181), citing many literatures. The Ra formula is well known, just like the delta E.
2) Can you explain the negative values for the average roughness reported in table 1? Did you mean to report relative values to a standard? Or those are absolute values? Can you show any negative roughness reported in the literature?
3) Line 188 is unnecessary, since the heading of the table 1 (line 187) already spells out that acronym.
4) Lines 262-269. What experimental techniques could you use to quantify the abrasive content? If researches don’t do that, conclusions will remain speculative. One suggestion is to partner with other departments in exact sciences/engineering for this task.
5) Table 3. While 3 toothpastes decreased their delta E values from S to L protocol, only the CBW toothpaste increased. What could have caused that? Can you correlate that finding with anything in its composition listed in table A2? The explanation given in lines 315-318 is not very clear.
Author Response
Dear reviewer,
Thank you very much for taking the time to review this manuscript and we appreciate the opportunity to clarify and improve the manuscript in line with your suggestions. Hereby the authors present a point-by-point response to all the comments and suggestions kindly given by you.
The corresponding revisions are marked in colored text (yellow) in the manuscript file (version R1). New references were added.
We hope the answers below and modifications introduced in the manuscript are clear and concise enough and that the manuscript now matches the high standards of the “New Techniques, Materials and Technologies in Dentistry: Second Edition”, a special issue of Applied Sciences that belongs to the section "Applied Dentistry and Oral Sciences”.
Your insightful feedback in the materials and methods section has contributed to an improvement in the study's design.
Comment 1: Please include the formula for the Surface roughness measurement (Ra) (line 161). The authors did include for delta E (line 181), citing many literatures. The Ra formula is well known, just like the delta E.
Response 1: Thank you for your feedback and for highlighting the importance of including the formula for surface roughness measurement (Ra). Recognizing its significance in surface characterization, we have incorporated the Ra formula:
(Due to formating issues the formula couldn't be inserted here in the 'Notes to Reviewer', it has been marked in yellow in the article PDF in the attachments)
where Ra represents the average roughness, L denotes the evaluation length of the roughness profile, and Z(x) signifies the deviation of the roughness profile from the mean line. This addition can be found in the revised manuscript at line 162.
References:
- De Oliveira, R.R.L.; Albuquerque, D.A.C.; Cruz, T.G.S.; Yamaji, F.M.; Leite, F.L. Measurement of the Nanoscale Roughness by Atomic Force Microscopy: Basic Principles and Applications. In Atomic Force Microscopy - Imaging, Measuring and Manipulating Surfaces at the Atomic Scale; InTech, 2012 ISBN 978-953-51-0414-8.
- Anderson, L. N., Alsahafi, T., Clark, W. A., Felton, D., & Sulaiman, T. A. (2023). Evaluation of surface roughness of differently manufactured denture base materials. The Journal of prosthetic dentistry, S0022-3913(23)00568-1. Advance online publication. https://doi.org/10.1016/j.prosdent.2023.08.028
Comment 2: Can you explain the negative values for the average roughness reported in table 1? Did you mean to report relative values to a standard? Or those are absolute values? Can you show any negative roughness reported in the literature?
Response 2: The authors appreciate the reviewer's inquiry regarding the negative values observed for the average roughness reported in table 1 and understand the comment. However, apologize for any confusion caused by this discrepancy. It's important to clarify that the values presented in table 1 are not indicative of average roughness (Ra), but rather represent surface roughness difference (ΔRa), the final value is the average difference between the final and initial values. To avoid confusion and ensure clarity for readers, authors have added “Δ” in table 2, consistent with the legend. This modification should facilitate a clearer understanding of the data. We apologize for any confusion this oversight may have caused and appreciate the opportunity to rectify it.
Comment 3: Line 188 is unnecessary, since the heading of the table 1 (line 187) already spells out that acronym.
Response 3: Thank you for your attention to detail and for highlighting the redundancy in line 188 regarding the acronym “NBS” mentioned in table 1. The authors acknowledge this redundancy and will remove the heading of the table 1 in line 188 to avoid duplication.
Comment 4: Lines 262-269. What experimental techniques could you use to quantify the abrasive content? If researches don’t do that, conclusions will remain speculative. One suggestion is to partner with other departments in exact sciences/engineering for this task.
Response 4: Thank you for your insightful questions regarding the quantify of abrasive content. In response, our research group has planned future studies involving toothpastes containing different whitening technologies. These studies will incorporate advanced experimental techniques such as Scanning Electron Microscopy (SEM) and Energy-Dispersive X-ray Spectroscopy (EDS) to accurately quantify abrasive content in the whitening toothpastes. Despite our efforts to obtain information directly from the brand, we encounter challenges in securing this data.
To clarify our approach in addressing this issue, the authors have added a paragraph to lines 262-266 in the revised article.
References:
- Hodoroaba, V.-D. Energy-Dispersive X-Ray Spectroscopy (EDS). In Characterization of Nanoparticles; Elsevier, 2020; pp. 397–417 ISBN 978-0-12-814182-3.
- Peetsch, A.; Epple, M. Characterization of the Solid Components of Three Desensitizing Toothpastes and a Mouth Wash. Materialwissenschaft Werkst 2011, 42, 131–135, doi:10.1002/mawe.201100744.
Comment 5: Table 3. While 3 toothpastes decreased their delta E values from S to L protocol, only the CBW toothpaste increased. What could have caused that? Can you correlate that finding with anything in its composition listed in table A2? The explanation given in lines 315-318 is not very clear.
Response 5: The authors understand the reviewer's comment and we decided to eliminate two paragraphs and added a commentary (see line 307-309) for the following explanation to make sense. In the lines 302-312, authors only wanted to gauge something about the difference between the two protocols. Although we know that this was not the primary focus of the study, nor was it subjected to statistical analysis, when looking at the results of protocol L, it was supposed the values to align or potentially exacerbate those recorded in protocol S, because Protocol L aimed to simulate the long-term effects of different toothpastes. The experimental conditions of Protocol L may have influenced the observed outcomes. As mentioned in the paragraph, the amount of toothpaste used in each brushing cycle of Protocol L replicated only that of a single brushing, which may not have provided sufficient exposure to the beneficial properties of the toothpaste over the course of the longer duration of the protocol. This could have affected the efficacy of certain toothpaste formulations in achieving desired outcomes such as enamel surface roughness and color assessment.
We acknowledge the reviewer's feedback regarding the clarity of the explanation and will ensure that the revised manuscript provides a more comprehensive and detailed account of the potential factors contributing to the observed differences in enamel surface roughness and ΔE values between the two protocols.
